https://doi.org/10.1038/s42003-021-02250-7　　**OPEN**
# Structural basis of coronavirus E protein interactions with human PALS1 PDZ domain

Airah Javorsky[1], Patrick O. Humbert [1,2,3,4✉] & Marc Kvansakul [1,2✉]

SARS-CoV-2 infection leads to coronavirus disease 2019 (COVID-19), which is associated with severe and life-threatening pneumonia and respiratory failure. However, the molecular basis of these symptoms remains unclear. SARS-CoV-1 E protein interferes with control of cell polarity and cell-cell junction integrity in human epithelial cells by binding to the PALS1 PDZ domain, a key component of the Crumbs polarity complex. We show that C-terminal PDZ binding motifs of SARS-CoV-1 and SARS-CoV-2 E proteins bind the PALS1 PDZ domain with 29.6 and 22.8 μM affinity, whereas the related sequence from MERS-CoV did not bind. We then determined crystal structures of PALS1 PDZ domain bound to both SARS-CoV-1 and SARS-CoV-2 E protein PDZ binding motifs. Our findings establish the structural basis for SARS-CoV-1/2 mediated subversion of Crumbs polarity signalling and serve as a platform for the development of small molecule inhibitors to suppress SARS-CoV-1/2 mediated disruption of polarity signalling in epithelial cells.

[1] Department of Biochemistry & Genetics, La Trobe Institute for Molecular Science, La Trobe University, Melbourne, Vic, Australia. [2] Research Centre for Molecular Cancer Prevention, La Trobe University, Melbourne, Vic, Australia. [3] Department of Biochemistry & Molecular Biology, University of Melbourne, Melbourne, Vic, Australia. [4] Department of Clinical Pathology, University of Melbourne, Melbourne, Vic, Australia. ✉email: p.humbert@latrobe.edu.au; m.kvansakul@latrobe.edu.au

The emergence of the novel coronavirus severe acute respiratory syndrome coronavirus 2 (SARS-CoV-2) at the end of 2019 in China led to a global pandemic with currently 120 M infections and over 2.5 M deaths[1]. In addition to the substantial morbidity and mortality associated with SARS-CoV-2 infection[2], the pandemic has led to major global economic dislocations, spurring a concerted worldwide campaign to develop therapeutics targeting this virus.

Whilst a substantial part of the global effort was aimed at the SARS-CoV-2 spike protein for vaccine development[3], other proteins have also been examined in detail[4]. Coronaviruses including SARS-CoV-2, the causative agent of COVID-19, comprise four major structural proteins, spike protein (S), nucleocapsid protein (N), membrane protein (M) and envelope protein (E)[5]. E protein is a 75–109 amino acid integral membrane protein and has been predicted to feature a short 7–12 residue hydrophilic N-terminal domain followed by a hydrophobic transmembrane domain of ~25 residues and a long hydrophilic C-terminal domain[6]. In addition to SARS-CoV-2, several other coronaviruses have been shown to be important human pathogens, including the closely related SARS-CoV-1 and MERS-CoV. All three are members of the betacoronavirus family, with SARS-CoV-1 and 2 belonging to clade b, whereas MERS-CoV belongs to clade c[7].

Functionally, the SARS-CoV-1 E protein has been shown to play a role in virus pathogenesis and may act as a virulence factor[8,9]. NMR studies revealed that peptides covering the central transmembrane region of SARS-CoV-1 E protein oligomerize in lyso-myristoyl phosphatidylglycerol (LMPG) micelles to form pentameric ion channels[10,11], whereas the C-terminal tail was shown to harbour a binding motif sequence that allows binding to PDZ (postsynaptic density protein (PSD95), Drosophila disc large tumor suppressor (Dlg1), and zonula occludens-1 protein (zo-1)) domains, an 80–90 amino acid structural domain commonly found in signalling proteins[12]. Importantly, the SARS-CoV-1 E protein's PDZ-binding motif[13] sequence allows interaction with the host cell polarity signalling protein PALS1 (Protein associated with Lin7/MPP5) thus interfering with epithelial function and integrity, a property of the SARS-CoV-1 E protein which is thought to be important for its virulence[13].

Cell polarity refers to the asymmetric distribution of cellular components including proteins, lipids and carbohydrates, and is a critical process to control animal tissue organization and development[14]. Apico-basal cell polarity is a common form of epithelial polarity that allows spatial segregation of cellular components between the apical and basolateral membrane and is also key for the establishment and maintenance of epithelial junctions[15]. One of the major complexes that regulate apicobasal cell polarity is the Crumbs (CRB) complex (Fig. 1a)[16,17]. The CRB complex consists of CRB (CRB1–3 in humans), PATJ (Pals1-associated tight junction protein) and PALS1[16]. PALS1 is a member of the MAGUK family (membrane-associated guanylate kinase) and is a signalling adaptor protein that consists of a number of protein-protein interaction domains including two L27 domains, a PDZ domain, an Src homology 3 (SH3) domain, a Hook domain, and a Guanylate Kinase (Guk) domain[16]. PALS1 regulates cell polarity through binding to the PDZ-binding domain (PBM) of CRB3 with loss of PALS1 leading to disruption of cell polarity including loss of integrity of tight and adherens junctions[18–23]. The CRB3 protein is the major crumbs isoforms expressed in human epithelial cells with the CRB3A splice form encoding the carboxy-terminal PBM sequence ERLI that enables PALS1 binding[24].

Importantly, in addition to being the binding site for the Crumbs PBM, the PDZ domain of PALS1 is also the target of certain viral proteins during viral infections[25]. Indeed, the SARS-CoV-1 E protein features a type II PDZ-binding motif (PBM)

harbouring the X-φ-X-φ$_{COOH}$ consensus sequence (where X represents any amino acid and φ is a hydrophobic residue such as V, I or L)[26] that has been shown to bind to PALS1 PDZ[13]. Following viral infection, PALS1 is recruited to virus-assembly sites at the Golgi where it co-localises with the SARS-CoV-1 E protein[13]. This interaction was shown to be dependent on the PBM of SARS-CoV-1 E protein, which competes with the CRB3 PBM for the PALS1 interaction (Fig. 1a), leading to disrupted localisation of PALS1, altered cell shape and disrupted tight junction assembly of epithelial cells[13]. Conversely, deletion of the PBM in SARS-CoV-1 E protein decreases expression of inflammatory cytokines and resulted in substantial virus attenuation[8,27], suggesting that E protein plays a prominent role in viral pathogenesis. Hijacking of PALS1 by SARS-CoV-1 E protein in infected lung cells is therefore postulated to lead to the loss of epithelial barrier function resulting in enhanced infiltration of SARS-CoV-1 virions into underlying tissues, thus contributing to the severe alveolar damages observed in post-mortem lung biopsies from SARS-CoV-1-infected patients[13]. However, the structural basis of E protein engagement of human PALS1 PDZ remains to be clarified.

We now report interaction and crystallographic studies examining the binding of Coronavirus E protein PDZ-binding motifs to the human PALS1 PDZ domain. Our results provide a structural basis for the subversion of Crumbs signalling by SARS-CoV-1 and -2, and provide a platform for the development of small molecule antagonists to exploit this for therapeutic intervention.

## Results and discussion

To understand the impact of the Coronavirus E protein binding to PALS1 on polarity signalling, we examined the affinity of recombinant human PALS1 PDZ (Fig. 1b) for 8-mer peptides spanning the C-terminal PBM of SARS-CoV-1_E, SARS-CoV-2_E and MERS-CoV_E (Fig. 1c). Our ITC measurements revealed that whilst PALS1 PDZ did not bind to MERS-CoV_E, it bound to SARS-CoV-1_E with an affinity of 29.6 μM, and SARS-CoV-2_E with an affinity of 22.8 μM (Fig. 1d). Our measurements differ from previously published findings that revealed an affinity of 130 μM for SARS-CoV-1_E and 40 μM for SARS-CoV-2_E[28], however, we note that our study used 8-mer peptides whereas Toto et al. used 10-mer peptides together with the PALS1 PDZ F318W mutant. Analysis of the thermodynamic parameters of SARS-CoV-1/2_E PBM binding to PALS1 PDZ revealed an unfavourable entropy change, with the interaction being driven by enthalpy (Table 1).

To understand the structural basis of the PALS1 PDZ and SARS-CoV-1/2 E PBM interaction, we determined the crystal structures of PALS1 PDZ bound to the SARS-CoV-1_E (SRVPDLLV) and SARS-CoV-2_E (EGVPDLLV) PBM peptides to resolutions of 1.74 and 1.9 Å, respectively (Table 2). As previously shown the PALS1 PDZ domains adopts a compact globular fold comprising five to six β-strands and two α-helices that form a β-sandwich structure[23,29]. Examination of both the PALS1 PDZ:SARS-CoV-1_E and PALS1 PDZ:SARS-CoV-2_E complexes showed that the E protein PBM peptides are bound in the canonical ligand-binding groove formed by the β2 strand and helix α2 (Fig. 2b–e). PALS1 PDZ:SARS-CoV-1_E and PALS1 PDZ:SARS-CoV-2_E superimposed with one another yielded an overall root-mean-square deviation (RMSD) of 0.72 Å (Fig. 2f). There was no discernible difference between PALS1 PDZ domains from the PDZ:SARS-CoV-2_E complex and PALS1 PDZ from an endogenous complex PALS1 PDZ:Crumbs (PDB ID 4UU5), which were superimposed to produce an overall RMSD of 0.80 Å (Fig. 2g).

In the PALS1 PDZ:SARS-CoV-1_E complex, SARS-CoV-1_E Val75 is docked in the conserved hydrophobic pocket formed by

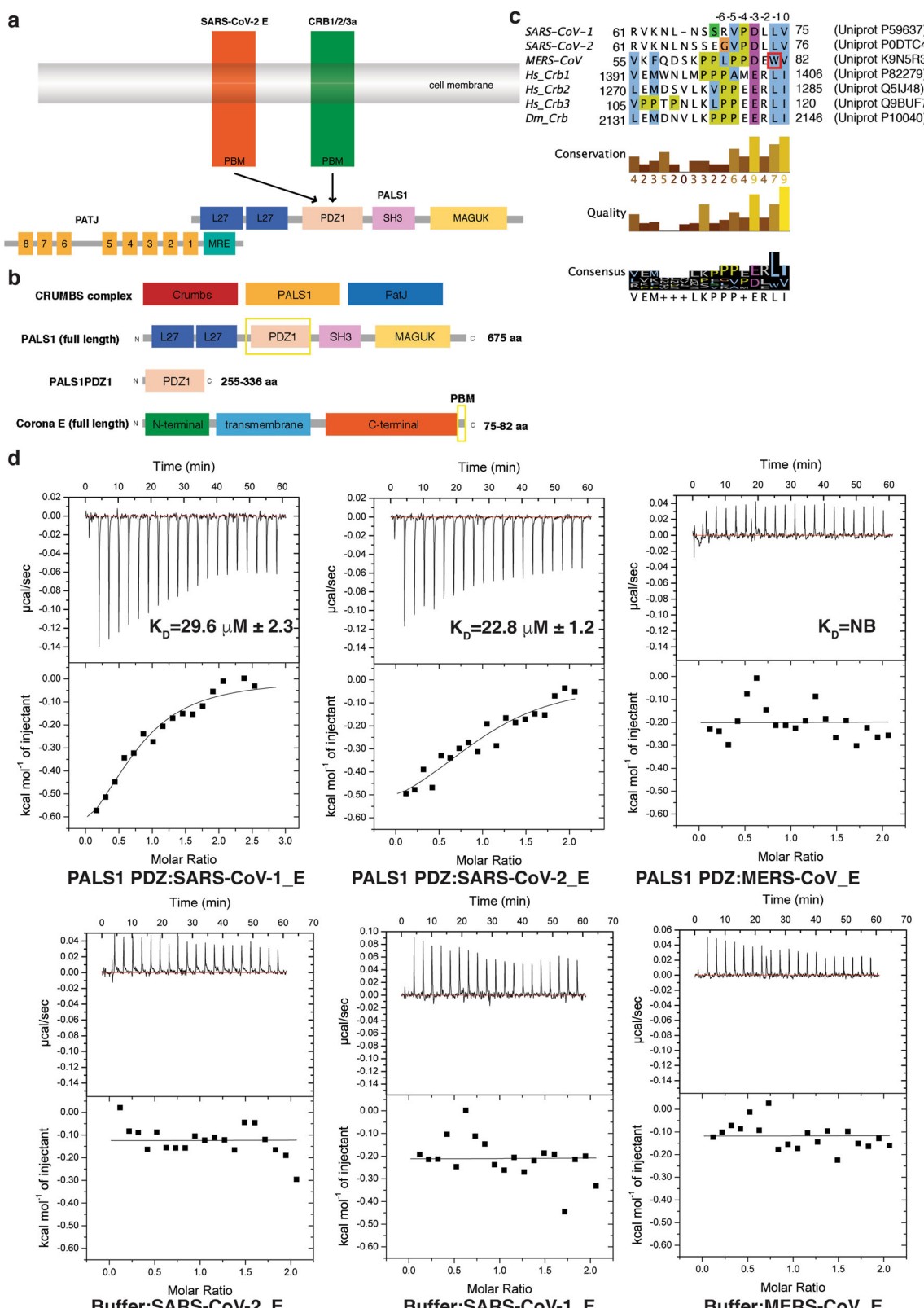

PALS1 PDZ Leu267, Gly268 and Ala269 (Fig. 3a). In addition, two hydrogen bonds are formed by Val271$^{PALS PDZ}$–Leu74$^{SARS-CoV-1\_E}$ located at the second β-sheet of PALS1 PDZ (Fig. 3a). Lastly, an ionic interaction is found between Arg272$^{PALS1 PDZ}$–Asp73$^{SARS-CoV-1\_E}$ (Fig. 3a), and SARS-CoV-2_E Val75, Leu74 and Leu 73 are involved in hydrophobic interactions.

Key hydrogen bond and ionic interactions between the PALS1 PDZ:SARS-CoV-2_E are identical with SARS-CoV-1_E (Fig. 3b). Major differences are found among the hydrophobic contacts, with the rotamer of SARS-CoV-1_E Leu73 differing from SARS-CoV-2_E Leu74. Furthermore, in SARS-CoV-1_E only Val75, Leu74 and Leu 73 are involved in hydrophobic contacts, whereas

**Fig. 1 Interactions of PALS1 PDZ domain with SARS-CoV-1, SARS-CoV-2 and MERS-CoV E protein PBM. a** Schematic outline of Crumb's complex interactions. **b** Domain organization of PALS1 and SARS-CoV-2 E protein. **c** Sequence alignment of C-terminal PBM motifs of coronavirus E proteins with human (hs) crumbs isoforms and *D. melanogaster (dm)* crumbs. Trp81 in the MERS CoV E protein is boxed in red. PBM sequences are numbered above the alignment with the first C-terminal residue designated as 0, with preceding residues numbered −1, −2, −3 etc. Uniprot accession numbers and sequence boundaries are listed for all sequences. Residue colouring is as follows: blue for hydrophobic, magenta for a negative charge, red for positive charge, yellow for prolines and white for unconserved. Three different conservation metrics calculated by Jalview are shown. Quality score is calculated for each column in an alignment by summing, for all mutations, the ratio of the two BLOSUM 62 scores for a mutation pair and each residue's conserved BLOSUM62 score, and then plotted on a scale from 0 to 1. The consensus displayed below the alignment is the percentage of the modal residue per column. The alignment was generated using Clustal Omega[44] followed by analysis using Jalview[45]. **d** Binding profiles of isolated PALS1 PDZ domain interaction with coronavirus E protein PBM peptides are displayed. Each profile is represented by a raw thermogram (top panel) and a binding isotherm fitted with a one-site binding model (bottom panels). $K_D$ dissociation constant, ± standard deviation, NB no binding. Each of the values was calculated from at least three independent experiments. Top panels show peptide into protein titrations, bottom panels show peptide into buffer control titrations.

---

**Table 1 Summary of affinity and thermodynamic binding parameters for PALS1 PDZ domain interactions with SARS CoV-1 and -2 PBM peptides.**

|  | $K_D$ (μM) | $\Delta H$ (kcal/mol) | $\Delta G$ (kcal/mol) | $-T\Delta S$ (kcal/mol/K) | $N$ |
|---|---|---|---|---|---|
| SARS-CoV-1 PBM | 29.6 ± 2.3 | −0.67 ± 0.01 | −6.15 ± 0.03 | 5.49 ± 0.04 | 1.0 ± 0.02 |
| SARS-CoV-2 PBM | 22.8 ± 1.2 | −0.68 ± 0.01 | −6.34 ± 0.02 | 5.66 ± 0.15 | 0.8 ± 0.1 |
| MERS-CoV PBM | NB | NB | NB | NB | NB |

NB denotes no binding. Each of the value was calculated from at least three independent experiments. Errors are ±SD.

---

**Table 2 X-ray data collection and refinement statistics.**

|  | PALS1 PDZ:SARS-CoV-1_E | PALS1 PDZ:SARS-CoV-2_E |
|---|---|---|
| *Data collection* |  |  |
| Space group | P 1 | P $2_1$ |
| No. of molecules in AU | 2 | 4 |
| *Cell dimensions* |  |  |
| *a, b, c* (Å) | 28.15, 39.96, 40.83 | 28.69, 103.76, 59.08 |
| *α, β, γ* (°) | 94.90, 108.78, 100.84 | 90.00, 103.98, 90.00 |
| Wavelength (Å) | 0.9537 | 0.9537 |
| Resolution (Å)* | 38.18-1.74 (1.8-1.74) | 27.84-1.9 (1.94-1.9) |
| $R_{sym}$ or $R_{merge}$* | 0.040 (0.150) | 0.129 (0.725) |
| $I / \sigma I$* | 12.9 (4.4) | 3.4 (0.8) |
| CC(1/2) | 0.998 (0.965) | 0.987 (0.431) |
| Completeness (%)* | 96.4 (90.5) | 99.4 (98.6) |
| Redundancy* | 3.6 (3.4) | 2.9 (2.9) |
| Wilson B-factor | 14.62 | 19.33 |
| *Refinement* |  |  |
| Resolution (Å) | 38.18-1.736 | 27.84-1.9 |
| No. of reflections | 16,267 | 26,171 |
| $R_{work}/R_{free}$ | 0.162/0.188 | 0.196/0.229 |
| *No. of non-hydrogen atoms* |  |  |
| Protein | 1378 | 2763 |
| Ligand/ion | 0 | 33 |
| Water | 306 | 359 |
| *B-factors* |  |  |
| Protein | 19.66 | 25.55 |
| Ligand/ion | 0 | 45.71 |
| Water | 31.01 | 30.33 |
| *R.m.s. deviations* |  |  |
| Bond lengths (Å) | 0.003 | 0.006 |
| Bond angle (°) | 0.61 | 0.59 |
| *Ramachandran plot (%)* |  |  |
| Favoured | 98.29 | 98.87 |
| Allowed | 1.71 | 1.13 |
| Disallowed | 0 | 0 |

*Values in parentheses are for the highest resolution shell.

in SARS-CoV-2_E Pro72, Val71 are also involved in hydrophobic contacts. However, despite the sequence differences between SARS-CoV-1_E and SARS-CoV-2_E PBM in positions −5 and −6, affinities for PALS1 PDZ are comparable, indicating that positions 0, −1, −2 and −3 are required for CoV_E PBM interaction with PALS1 PDZ and define binding, whereas positions −4, −5, −6 and −8 do not play a major role. In contrast, CRB1 engagement of PALS1 PDZ differs from SARS-CoV_E binding due to notable differences in the CRB1 PBM sequence (Fig. 1c) such as the −2 position where Arg1404[CRB1] contacts α2 to form a hydrogen bond with Asn315 [PALS1 PDZ] and −3 position Glu1403 [CRB1] forms an ionic interaction and hydrogen bond to the β3 with Arg282[PALS1 PDZ] (Fig. 3c)[29]. The ionic interaction between Glu1403 [CRB1] and Arg282[PALS1 PDZ] mimics the interaction seen in the E protein complexes, however, due to the presence of the shorter side chain in Asp in E PBM compared to the Glu found in CRB1 the acceptor residue in PALS1 PDZ is Arg272 (Fig. 3c). It was previously shown that binding of CRB1 PBM to PALS1 required movement of the gatekeeper Phe318, which in the unbound form blocks access to the canonical ligand-binding groove in PALS1 PDZ[29]. A comparison of our structures with apo-PALS1 PDZ reveals that similarly, both SARS-CoV-1_E and SARS-CoV-2_E binding to PALS1 PDZ trigger rotation of PALS1 Phe318 out of the ligand-binding groove to allow binding of SARS-CoV-1/2_E PBM (Fig. 3d).

A comparison of MERS-CoV E PBM with SARS-CoV-1_E and SARS-CoV-2_E reveals that the major difference is the presence of a large bulky Trp in MERS-CoV E PBM position −1 (Fig. 1b), where SARS-CoV-1_E and SARS-CoV-2_E, as well as Crumbs, feature the smaller Leu. With MERS-CoV E PBM displaying no detectable affinity for PALS1 PDZ we surmise that the Trp in the −1 position is too bulky to be accommodated in the PALS1 PDZ binding groove and particularly the pocket accommodating the −1 Leu from Crumbs PBM (Fig. 3e), thus completely ablating the interaction.

A 6-mer peptide spanning the CRB1 PBM motif bound to PALS1 PDZ with an affinity of 16.8 μM[29], comparable to the affinities we determined for our E protein PBM 8-mer peptides. This suggests that CoV_E proteins do not outcompete CRB from

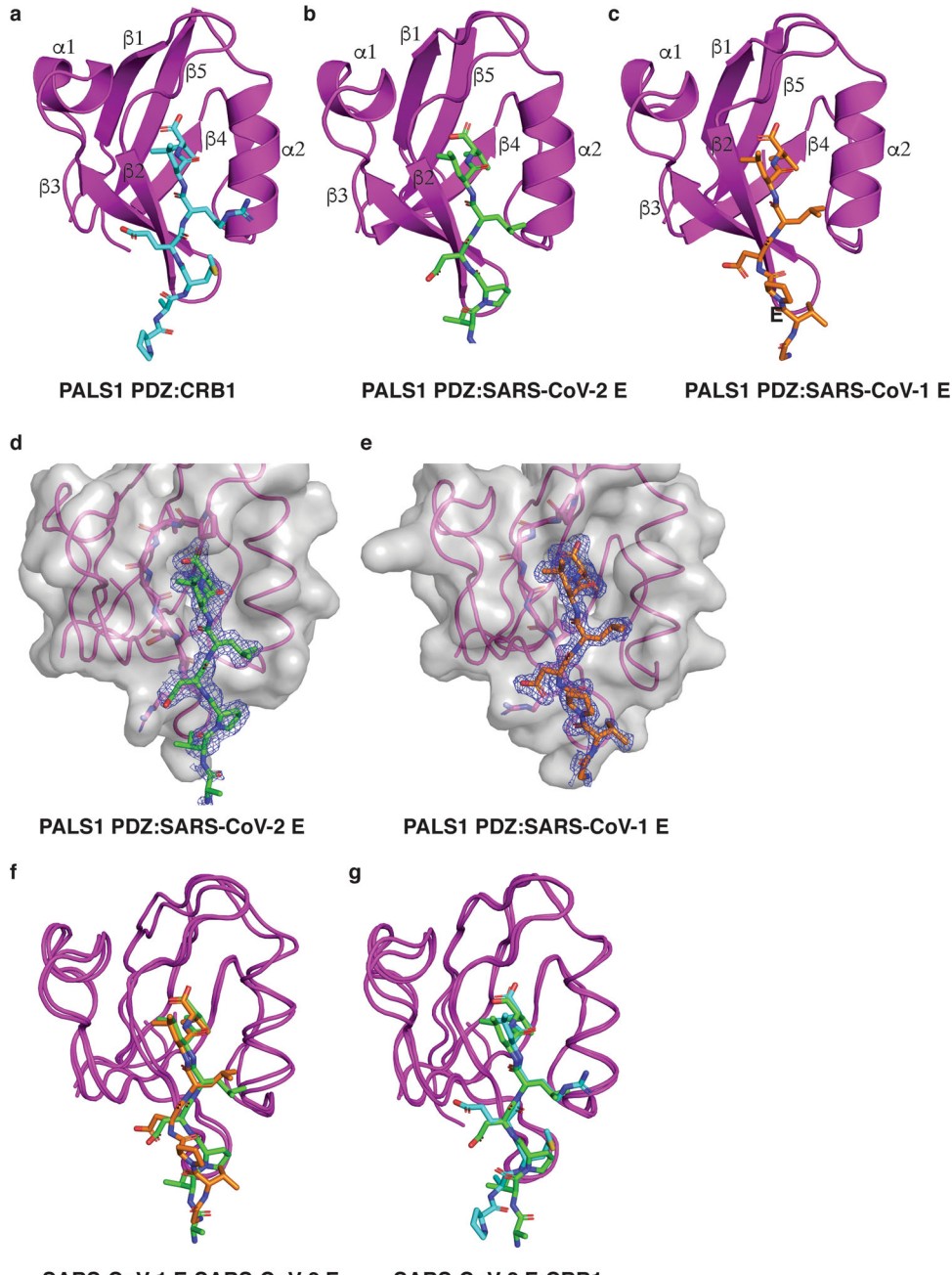

**Fig. 2 The crystal structures of PALS1 PDZ bound to SARS-CoV-1 and SARS-CoV-2 E protein PBM.** The E protein PBM peptide engages PALS1 PDZ via a shallow groove located between the β2 and α2. **a** Human PALS1 PDZ (magenta) is shown as a cartoon with a human CRB1 peptide (cyan) represented as sticks (PDB ID 4UU5)[29]. **b** PALS1 PDZ (magenta) is shown as a cartoon with SARS-CoV-2 E protein PBM (green) represented as sticks. **c** PALS1 PDZ (magenta) is shown as a cartoon with SARS-CoV-1 E protein PBM (orange) represented as sticks. **d, e** Electron density maps centred on the SARS-CoV-2 E protein PBM peptide (green) or SARS-CoV-1 protein PBM peptide (orange) in the binding groove of PALS1 PDZ. The electron density map is a 2Fo–Fc map shown as a blue mesh contoured at 1.5σ. **f** Overlay of ribbon traces of PALS1 PDZ with SARS-CoV-2 E protein PBM (green) and PALS1 PDZ with SARS-CoV-1 E protein PBM peptide (orange). **g** Overlay of ribbon traces of PALS1 PDZ with SARS-CoV-2 E protein PBM (green) and human Crumbs peptide (cyan) peptide (PDB ID 4UU5)[29].

PALS1, and that the effect of perturbed polarity signalling is due to the forced mislocalization of PALS1 after binding to CoV_E. In SARS-CoV-1 infected cells, PALS1 is specifically recruited to the virus budding compartment (ERGIC/Golgi region)[13]. As correct localisation of cell polarity proteins is essential for their function, mislocalisation of PALS1 would thus result in the altered cell shape and disrupted tight junction assembly of epithelial cells observed following SARS-CoV-1infection of epithelial cells[13]. Interestingly, mislocalisation is specific to PALS1 as another PDZ domain-containing protein, ZO-1 remained localised to tight junctions[13]. The differential binding of MERS vs. SARS-CoV-1/2 is also consistent with differential virulence. Interestingly, SARS-CoV infects polarised epithelium specifically through the apical cell membrane[30], whereas MERS enters both through apical and basolateral membranes, and viral budding occurs via apoptosis[31], which raises the possibility that the differences in viral entry could also reflect the differential ability to perturb polarity signalling. As PALS1 is also expressed in endothelial cells where it

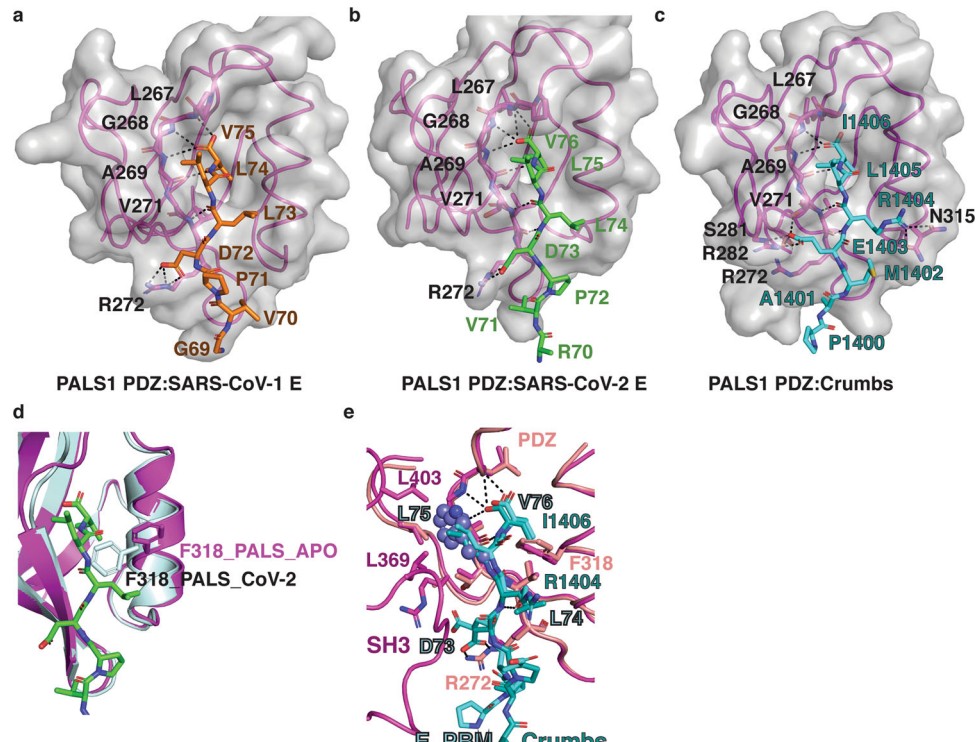

**Fig. 3 Detailed view into the interfaces of PALS 1 PDZ bound to SARS-CoV-1 and SARS-CoV-2 E protein PBM. a–c** Detailed view of interactions in the binding groove of PALS1 PDZ in complex with **a, b** SARS-CoV-2 E protein PBM peptide (green) or SARS-CoV-1 E protein PBM peptide (orange) or **c** Crumbs PBM peptide (cyan) (PDB ID 4UU5)[29]. Residues involved in hydrogen bonds or ionic interactions (shown as black dotted lines) are labelled. **d** Superimposition of PALS1 PDZ:SARS-CoV-2_E_PBM (magenta cartoon and green sticks) and apo-PALS1 PDZ (light blue). The orientation of the gatekeeper F318 is shown as sticks. **e** Superimposition of PALS1 PDZ:SARS-CoV-2_E_PBM with PALS-PDZ-SH3-MAGUK:Crumbs_PBM complex. PALS1 PDZ:SARS-CoV-2_E_PBM is shown as pink cartoon and sticks or cyan sticks, respectively. PALS-PDZ-SH3-MAGUK (PDB ID 4WSI)[23] is shown as light pink cartoon and sticks. Dm Crumbs_PBM is shown as dark cyan sticks. Key residues are labelled, and key hydrogen or ionic bonds are denoted by black dashed lines. The modelled Trp 81 from MERS CoV E PBM is shown as steel blue spheres.

regulates vasculature morphogenesis through binding and modulation of VE–cadherin function[32], CoV_E interference with PALS1 function could also play a direct role in compromising endothelial cell–cell junctions and inducing vascular leakage observed in SARS-CoV-2 patients[33]. Of note, VE–cadherin levels are also downregulated as a result of COVID-19 infection[34] suggesting endothelial cells could be already sensitised to further disruption of junctional regulators.

Subversion of polarity signalling is a feature associated with a number of viral infections, including those by human papillomavirus (HPV), human T lymphotropic virus-1 (HTLV), Tick-Borne encephalitis virus (TBEV) and influenza A virus (IAV)[25]. For tumorigenic viruses such as HPV, the ability to manipulate host cell polarity pathways has been shown to be strongly associated with the development of subsequent malignancies, whereas for non-malignant viruses including TBEV and IAV subversion of polarity signalling is associated with increased virulence and evasion of immune defences[25]. Evidence for SARS-CoV-1_E suggests that similar to other non-malignant viruses the ability to modulate polarity signalling is associated with increased virulence[13,27].

In summary, we have shown that SARS-CoV-1_E and SARS-CoV-2_E PBM bind to PALS1 PDZ with comparable affinities, and established the structural basis for CoV_E PBM binding to PALS1 PDZ. These findings provide a mechanistic basis for coronavirus induced perturbation of polarity signalling and may serve as a platform for the design of small-molecule inhibitors to

target the envelope protein interaction with PALS1 for antiviral therapy.

## Methods

**Protein expression and purification**. The codon optimised cDNA encoding PALS1 (Uniprot accession number: Q8N3R9) PDZ domain (residues 255-336) was cloned into the bacterial expression vector pGex-6p-1 (GenScript). Target proteins were expressed using *Escherichia coli* condon+ in 2YT media (1.6% w/v Tryptone, 1.0% w/v yeast extract and 0.5% w/v sodium chloride (NaCl)) supplemented with 100 μg/ml ampicillin at 37 °C for 20 h. Protein expression was induced using the auto-induction protocol[35]. Cells were harvested by centrifugation at 4000$g$ for 20 min using the JLA 9.1000 rotor (Beckman Coulter Avanti J–E), then resuspended in 100 ml buffer A (50 mM Tris pH 8.0, 150 mM NaCl, 1 mm ethylenediaminetetraacetic acid (EDTA)). Lysis was carried out via sonication at the amplitude of 50 for 4 min with a 5-s gap after every 30-s sonication interval, using the Qsonica Q700 Sonicator. The resultant lysate was transferred into SS34 tubes for further centrifugation at 18,000 rpm for 30 min using the JA-25.50 rotor (Beckman Coulter Avanti J–E). The supernatant obtained after centrifugation was then filtered through a 0.45 μm pore size filter unit (Merck Millipore). The filtered supernatant was loaded onto 5 mL of glutathione sepharose 4B (GE Healthcare) equilibrated with buffer A. The column was washed with 100 ml of buffer A and protein on-column cleavage was achieved by adding 3 mg human rhinovirus 3 C protease and incubating for 16 h at 4 °C. 100 ml of buffer A was then added to collect cleaved protein, with the flowthrough concentrated at 4 °C using 3 kDa molecular weight cut-off centrifugal concentrator (Amicon® Ultra 15 Millipore) with a benchtop centrifuge (Allegra X-15, Beckman Coulter) at 4000$g$. PALS1 PDZ was subjected to size-exclusion chromatography using a Superdex S75 increase 10/300 column mounted on an ÄKTA Pure system (GE Healthcare) equilibrated in TRIS buffer (25 mM TRIS pH 8, 200 mM NaCl) or HEPES buffer (20 mM HEPES, 150 NaCl) and fractions were analysed using sodium dodecyl sulphate-polyacrylamide gel electrophoresis (SDS-PAGE). The final sample purity was estimated to be higher than 95% based on SDS-PAGE analysis. Appropriate fractions were collected and concentrated using a centrifugal concentrator with 3 kDa molecular weight cut-off (Amicon® Ultra 15) to a final

concentration of 1.5 mg/ml with protein in TRIS buffer subsequently used for ITC analysis, and at 3.5 mg/ml in HEPES buffer used for crystallisation. The concentration of proteins was quantified using a Nanodrop UV spectrophotometer (Thermo Scientific) at wavelength 205 nm via the Scopes method[36].

**Measurement of dissociation constants.** Binding affinities were measured for PALS1 PDZ in TRIS buffer at a final concentration of 90 µM. Peptide ligands were equilibrated in TRIS buffer and used at a concentration of 900 µM, and all affinity measurements were performed in triplicate. Peptides of E protein used were SARS-CoV-1: SRVPDLLV (Uniprot accession code P59637, residues 68–75) (Mimotopes), SARS-CoV-2: EGVPDLLV (UniProt accession code P0DTC4, residues 69–76) (Mimotopes) and MERS-CoV: PLPPDEWV (UniProt accession code K9N5R3, residues 75–82) (Mimotopes). Peptides dissolved in water to prepare 5 mM stock solutions and subsequently diluted with buffer for the final experiment. ITC experiments were conducted using a MicroCal iTC200 System (GE Healthcare) at 25 °C with a stirring speed of 750 rpm using 20 injections as previously described[37]. Specifically, we used an initial 0.4 µL injection followed by nineteen 2.0 µL injections to determine affinities and thermodynamic parameters. Data were analysed using the Origin 7.0 software (OriginLab Corporation) using a one-site binding model.

**Crystallisation and structure determination.** Complexes of PALS1 PDZ domain in HEPES buffer with coronavirus peptides were reconstituted by mixing protein and peptide were mixed at a molar ratio of 1:4 and then concentrated to 4 mg/ml using 3 kDa cut-off (Amicon® Ultra 0.5) at 4 °C. Concentrated protein complex samples were subjected to high-throughput crystallisation screening using a Gryphon LCP (Art Robbins Instruments) with 0.2 µl of protein sample and 0.2 µl reservoir solution per drop using 96-well sitting drop trays (swissic, Neuheim, Switzerland) and a range of commercial screens using the sitting drop vapour diffusion method at 20 °C in-house.

Crystals were mounted on nylon and copper loops (MiTGen). All data were collected at the Australian Synchrotron using the MX2 beamline equipped with the Eiger 16 M detector (Dectris) with an oscillation range of 0.1° per frame with a wavelength of 0.9537, integrated using XDS[38] and scaled using AIMLESS[39].

PALS1 PDZ:SARS-CoV-1_E crystals were grown in 0.1 M Bis-Tris pH 6.5, 25 % w/v PEG 3350 and flash cooled at −173 °C in the mother liquor. The PALS1 PDZ:SARS-CoV-1 complex formed plate-shaped crystals belonging to space group P1 with $a = 28.15$ Å, $b = 39.96$ Å, $c = 40.83$ Å, $\alpha = 94.90°$, $\beta = 108.78°$, $\gamma = 100.84°$. Molecular replacement was carried out using PHASER[40] with the previously solved structure of apo PALS1 PDZ domain (PDB ID: 4UU6)[29] as a search model. PALS1 PDZ:SARS-CoV-1_E crystals contained two molecules each of PALS1 PDZ and SARS-CoV-1 peptide in the asymmetric unit, with 43.5% solvent content. The final TFZ and LLG values after molecular replacement were 11.9 and 113.72, respectively. The final model was built manually over several cycles using Coot[41] and refined using PHENIX with a final Rwork/Rfree of 0.16/0.19, with 98.3% of residues in the favoured region of the Ramachandran plot and 0% of rotamer outliers.

PALS1 PDZ:SARS-CoV-2_E crystals were grown in 0.2 M Sodium acetate trihydrate, 0.1 M TRIS pH 8.5, 30% w/v PEG 4000 and flash cooled at −173 °C in mother liquor supplemented with 25% ethylene glycol. The PALS1 PDZ: SARS-CoV-2 complex formed prism-shaped crystals belonging to space group P2₁ with $a = 28.69$ Å, $b = 103.77$ Å, $c = 59.07$ Å, $\alpha = 90.00°$, $\beta = 103.97°$, $\gamma = 120°$. Molecular replacement was carried out using PHASER[40] with the previously solved structure of apo PALS1 PDZ domain (PDB ID: 4UU6)[29] as a search model. PALS1 PDZ:SARS-CoV-2_E crystals contained 4 molecules each of PALS1 PDZ and SARS-CoV-2 peptide in the asymmetric unit, with 47% solvent content. The final TFZ and LLG values after molecular replacement were 15.3 and 454.1, respectively. The final model was built manually over several cycles using Coot[41] and refined using PHENIX[42] with a final Rwork/Rfree of 0.19/0.23, with 99.4% of residues in the favoured region of the Ramachandran plot and 0% of rotamer outliers.

**Statistics and reproducibility.** All isothermal titration calorimetry experiments were performed in triplicates from discrete samples, with errors calculated as ±SD.

**Reporting summary.** Further information on research design is available in the Nature Research Reporting Summary linked to this article.

## Data availability

Data supporting the findings of this manuscript are available from the corresponding authors upon reasonable request. Coordinate files were deposited at the Protein Data Bank (https://www.rcsb.org/) using accession codes 7NTJ and 7NTK for PALS1 PDZ: SARS-CoV-1_E and PALS1 PDZ:SARS-CoV-2_E, respectively. The raw X-ray diffraction data were deposited at the SBGrid Data Bank[43] (https://data.sbgrid.org/data/) using their PDB accession code 7NTJ and 7NTK for PALS1 PDZ:SARS-CoV-1_E and PALS1 PDZ: SARS-CoV-2_E, respectively.

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

## Acknowledgements
We thank staff at the MX beamlines at the Australian Synchrotron for help with X-ray data collection. We thank the ACRF for their support of the Eiger MX detector at the Australian Synchrotron MX2 beamline and the Comprehensive Proteomics Platform at La Trobe University for core instrument support. This research was funded by La Trobe University (Scholarship to A.J.).

## Author contributions
A.J.: Experimental design, acquisition of data; analysis and interpretation of data; drafting and revising the article; P.O.H.: Conception and design; analysis and interpretation of data; drafting and revising the article; M.K.: Conception and design; acquisition of data; analysis and interpretation of data; drafting and revising the article.

## Competing interests
The authors declare no competing interests.
