## [Peer Review File · Communications Biology]

Reviewers' Comments:

Reviewer #1:

Remarks to the Author:

In 'Structural Basis of Coronavirus E protein interactions with human PALS1 PDZ domain', Javorsky and colleagues characterise the binding of the PDZ domain of Protein Associated with Lin7 (PALS1) in complex with PDZ-binding motif (PDM) of the E protein of SARS-CoV-1 and SARS-CoV-2. The authors use a biophysical approach, using calorimetry and X-ray crystallography. The main conclusion of this work is that the PDM of SARS-CoV-1 and SARS-CoV-2 bind with micromolar affinity to PALS1, interfering with the physiological association of this protein with members of an apical polarity complex, thus potentially disrupting the integrity of tight and adherens junctions.

The result herein presented are not entirely novel since that Teoh et al already described the interaction of SARS CoV-1 protein E with PALS1 (Mol Biol Cell. 2010 Nov 15;21(22):3838-52). However, the authors provide an expansion of this knowledge by characterising the interaction of SARS-CoV-2 and MERS-CoV with PALS1, particularly by presenting atomic resolution structures of the complexes formed by the PDM domains of the E proteins of the viruses and PALS1.

My opinion on the manuscript is that is very well written and structured, with overall well organised figures and clear description of the results and methods. However, I recommend its publication after amendment/clarification of few issues, listed below, identified in each section of the manuscript.

Introduction

The introduction is very well written and clear, covering all the relevant topics for the results presented. Few minor corrections are required:

1. In the first and third paragraphs, few acronyms are missing for the words SARS-CoV-2, LMPG, PDZ. In the third paragraph, the name of one of the proteins studied in this manuscript is presented, PALS1; however, the acronym is presented later in the introduction. It should be reported the first time the name is presented.
2. In the fourth paragraph, the three major complexes regulating apicobasal cell polarity are presented, CBR, PAR and SCRIB, but the rest of the manuscript focuses on PALS1, which is part of the CRB complex (acronym missing and to be added). Since PAR and SCRIB are not described, I think they should be removed to avoid confusion. I would rephrase into something like "One of the major complexes regulating apicobasal cell polarity is the the CRB complex...".
3. In the fifth paragraph, the Crumbs PDM is introduced, but no description is provided. It might be worth adding a sentence to describe, with a comprehensive reference.
4. A personal opinion more than a recommendation or a requested amendment, I think that a figure with a schematic of the different domains of SARS-CoV E protein, as well as the polarity complex, might help to visualise better the whole interactome presented in the manuscript and would really make the introduction outstanding.

Results and discussion

This section is clear, and the results interpretation is careful and well discussed.

However, I would like the authors to expand on few points, as well as amend the figures presented, as it follows:

1. In the first paragraph, the authors present the K_d measurements of interaction between PALS1 PDZ and SARS-CoV-1/2 E PBM peptides by ITC. Fig 1B present the ITC data, however the control titrations of the peptides into buffer are not reported. These should be added, either in the main figure or in a supplementary figure. This is particularly relevant because the heat of dilution observed at the end of the titration is quite high (almost half of the total signal recorded). This could be due to buffer mismatch (it is not clear from the Methods weather the peptides were resuspended in the very same protein buffer) or to some contaminants of the peptides synthesis that are titrated, generating heat (this can happen with commercially synthesised material). Also, the authors should report a table with averaged stoichiometry (n), ΔH, -TΔS (±SD or SEM)

obtained in the titrations. Ideally, considering that the PDZ and SARS-CoV-1/2 E PBM peptides complexes are very similar, similar enthalpy and entropy should be expected. If great differences were observed, the data should be discussed accordingly.

2. In the second paragraph, the authors introduce the structure of the complexes formed by PALS1 PDZ and SARS-CoV-1/2 E PBM peptides. The XRC structures show no folding of the small peptides. I was wondering whether the unstructured PBM peptides herein observed reflect previously reported structures of the SARS-CoV-1/2 E PBM (in greater constructs maybe) or PBM of other proteins. If the PBM is "usually" a structured element in greater constructs, the authors should discuss this. Another reason for investigating this more is that the high heat of dilution observed at the end of the ITC titration might be due to the transition of the peptides from (partially) folded to unfolded state during the binding. The analysis of the enthalpy and entropy obtained in the ITC experiments will guide this too.

3. In paragraph 4, the authors indicate the amino acids of the SARS-CoV-1/2 E PBM peptides with a negative numbering. My understanding is that position 0 is fixed at V75 (SARS-CoV-1_E) and V76 (SARS-CoV-2_E). Not having the numbering anywhere stated is though confusing, so I would report the numbering somewhere, maybe at the top of the alignment presented in Fig 1A. More on Fig 1A, the alignment reports a series of additional proteins that are never mentioned in the manuscript main text. I am assuming these are PBM of other proteins. The authors should clarify that.

Finally, this alignment is a tad "amatorial". A better alignment (with rigorous homology values) should be performed in Clustal (or similar), where it is possible to appreciate the conservation between PBM (all of those presented, not just the viral ones). A more detailed discussion on this figure should then be reported in the main text, since this is at the moment missing.

4. In paragraph 4, the authors describe the role of F318 in the binding of CRB1 PBM to PALS1 and how this reflects on the binding SARS-CoV-1_E and SARS-CoV-2_E binding to PALS1 PDZ. How did the authors extrapolate the information about the rotation of F318 in the interaction with SARS-CoV-1/2_E? Was this obtained through alignment between complexes and apo-PALS1 PDZ? A more detailed discussion on this should be provided and a panel in Fig 3 focused on the description of this important point should be presented.

5. In paragraph 5, the fact that MERS-CoV E does not bind to PALS1 PDZ is rationalised through the fact that it presents a W where SARS-CoV-1/2_E have a V. Please, point the Trp in the alignment of Fig 1A to help the reader.

Few minor amendments or suggestions:

1. Paragraph 2: In the sentence "Examination of both the PALS1 PDZ:SARS-CoV-1_E and PALS1 PDZ:SARS-CoV-2_E complex showed that the E protein PBM peptides are bound in the canonical ligand-binding groove formed by the β 2 strand and helix α 2 (Fig. 2)." 'Complex' should be amended into 'complexes'.

2. The Fig 3B is described before Fig 3A. It might be worth swapping 3B and 3A figures to avoid confusion for the readers.

3. In paragraphs 3 and 4, the amino acids of SARS-CoV-1/2_E interacting with PALS1 PDZ are reported in "random" order. I understand that this follows the top-to-bottom representation in Fig 3, however, at this level it might be better to follow an alphanumerical order. I will leave this to the authors.

4. A reference is missing for the sentence in paragraph 6 "A 6-mer peptide spanning the CRB1 PBM motif bound to PALS1 PDZ with an affinity of 16.8 μ M, comparable to the affinities we determined for our E protein PBM 8-mer peptides."

5. In the sentence "As correct localisation of cell polarity proteins are essential for their function, mislocalisation of PALS1...", 'are' should be corrected into 'is'.

6. In the sentence "In summary, we have shown that SARS-CoV-1_E and SARS-CoV-2_E PBM bind to PALS1 PDZ with comparable affinities and establish the structural basis for CoV_E PBM binding to PALS1 PDZ. These findings provide a mechanistic basis for coronavirus induced perturbation of polarity signalling and may serve as a platform for the design of small molecule inhibitors to target the envelope protein interaction with PALS1 for antiviral therapy" there are two typos: '..comparable affinities, and established...' and '..polarity signalling, and may...'.

Figures

Fig 1

See previous comments in the Results and Discussion section.

Note that the space between values and units are missing.

Fig 2

I think it would be beneficial if this figure was reorganised so that panel C becomes panel A, since that the PALS PDZ : CRB1 complex was not characterised in this work. Also, it might be worth indicating in this complex the structural elements that are interacting with the E proteins, mentioned in the paragraph 3 of the Results section, to guide the reader through the results.

Fig 3

1. Fig 3 reports 2 panels (D and E) that are never mentioned in the main text. These two panels should be moved to figure 2, since that they are actually reporting on the quality of the XRC electron density fitting.

2. I found Fig 3F quite busy, thus very difficult to follow. Few amendments are recommended. I would suggest focusing on the surface around V76 SARS-CoV-1/2_E only, since this is the object of the discussion here, and to align it with W81 of MERS-CoV_E to point at the different steric hindrance (I guess that the structure of MERS-CoV_E PBM could be easily modelled for this purpose). The surface is not necessary, so it could be probably hidden, so that the reader can focus on the V76/W81 vs PALS1 residues only. Also, there are way too many colours in this figure, making it quite complex to read. I would suggest to have each complex in the same colour, but two different shades for PALS1 and CoV_E protein respectively, so that it is easy to appreciate the interaction in each complex. Also, I believe that the SH3 domain of PALS-PDZ-SH3-MAGUK is not adding to this, but I will leave this to the author if they discuss better in the results the point of aligning this too.

Finally, a minor amendment. The PALS-PDZ-SH3-MAGUK:Crumbs_PBM complex is referred as PALS-PDZ-SH3-GUK:Crumbs_PBM complex in the caption. I am not sure whether this is a typo, if so, please correct.

Materials and methods

The methods are very well written, clear, and comprehensive. Rarely have I encountered such detailed description of the methods, particularly in papers where XRC data are presented, so I would really like to compliment the authors for the good job. Few minor corrections.

Protein expression and purification

1. *E. coli* should be presented in italic.

2. Note that some units are missing the space after the number (i.e. 3.5mg/mL instead of 3.5 mg/mL - this would apply to the whole manuscript).

Crystallisation and structure determination

1. In the first paragraph, the authors use the words "Corona peptides". A personal opinion, more than a recommendation, descriptions such as "Coronavirus peptides" or "viral peptides" sound more appropriate.

2. Still in the first paragraph, the authors describe the full set of plates tested in automated crystallisation trials. I think this is redundant, since that the successful crystallisation conditions are stated clearly in the third paragraph. I would suggest this is removed to avoid confusion for the reader. At the same time, if the authors observed crystals in different conditions tested and feel like sharing this information with the community, then they can provide more details.

Overall, I reiterate that I suggest the publication of this paper upon the suggested amendments, since that the scientific community would benefit of the new structures reported for drug discovery purposes.

Moreover, I wish the authors and their families health and some semblance of stability in these trying times.

Kind regards and stay safe.

Reviewer #2:

Remarks to the Author:

The authors present the crystal structures of SARS-CoV1 and SARS-CoV2 E proteins in complex with the PALS1 PDZ domain. MERS-CoV E was also tested although showed no binding affinity in iTC studies unlike the SARS-CoV1 and 2 E proteins which showed affinities of 29.6 and 22.8uM, respectively. The authors also compared the structures and interactions with that of PALS1 PDZ with its cellular binding partner, the Crumbs polarity complex (via a CRB1 peptide). It is hypothesised that the disruption of this interaction (through binding of the coronavirus E protein into the same pocket) is responsible for the increase in virulence observed between MERS-CoV and SARS-CoV 1 and 2 as well as the vascular leakage observed in some COVID-19 patients.

The findings presented are novel, convincing and propose potential targets for antiviral therapies that could assist in the fight against the COVID-19 pandemic as well as potential future outbreaks. The manuscript is succinct and to-the-point.

Comments:

- In the abstract sentence that reports the binding affinities (lines 32-33), the protein that was tested is not named (SARS-CoV1 and 2 E protein).
- The main explanation of binding on page 4 is well explained with sufficient, but not excessive, detail.
- Extension of 'Fig. 1' to Fig 1A, for example, may be helpful throughout the manuscript. Where this is done, panel B is often mentioned in the text before panel A. Perhaps panels A and B should be switched to show SARS-CoV1 results before SARS-CoV2 if they are referred to in that order.
- Isothermal titration calorimetry experiments seem to not have been taken fully to completion. To improve this, additional points using both less and more ligand could allow a complete sigmoidal curve to be generated. However, if this has proved challenging, the data remain valid as all points appear close to the fitted curve. This is supported by the binding affinity of PALS1 PDZ:SARS-CoV2 E presented here being close to that previously reported (although SARS-CoV1 E is quite different but this is addressed in the main text).
- The authors refer to certain secondary structure features (i.e. beta strands) although these are not annotated on any of the figures. The addition of a panel showing these (or annotation of a current figure) may aid the reader in identifying the regions being referred to in the text, although not essential.
- On line 124, the authors state that there is no significant difference between PALS1 PDZ:SARS-CoV2 E and PALS1 PDZ:Crumbs although figure 2E shows quite a large difference in the lower region of the E/Crumbs overlay. Perhaps the text should be altered to specify no significant difference in the upper/central regions or 'most' of the peptides.
- The comparison to MERS-CoV is well written and supported by the data.
- The abbreviation of human T lymphotropic virus-1 is missing in line 183.
- The sentence starting on line 210 doesn't make sense. Perhaps 'in which' needs re-wording on line 212?
- What is the composition of buffer A? First mentioned in line 214.
- Line 218 and 245: missing word 'concentrators' or similar after '...molecular weight cut off...'. I.e. '...molecular weight cut off' what?
- Ramachandran plots are good with over 98% of residues in the favoured region for both structures.
- Table showing data/resolution of the structures is presented., however, these are not mentioned in the text or figure legends.
- Figure 2 legend: proteins mentioned in A and B are the wrong way around compared to the labels in the figure panels. I.e. Legend states A shows SARS-CoV1 E although the figure is labelled as SARS-CoV2 E. The final line of the legend is also confusing and potentially contains an error in

line 467 where the Crumbs peptide is labelled as both cyan and forest green (with SARS-CoV2 E in green).

- Figure 3 legend is very repetitive and could be re-worded to be more succinct.

- Do panels 3D and E add anything to the figure? May be easier to see if shown without the surface of PALS1 PDZ. Panel F is good in that it shows the amino acid labels although appears very busy and difficult to interpret. Again, this may benefit from the removal of the surface view of PALS1 PDZ.

Reviewer #3:

None

We thank both reviewers for their detailed and helpful comments, and their overall positive views on our work.

Reviewer #1:

1. In the first and third paragraphs, few acronyms are missing for the words SARS-CoV-2, LMPG, PDZ. In the third paragraph, the name of one of the proteins studied in this manuscript is presented, PALS1; however, the acronym is presented later in the introduction. It should be reported the first time the name is presented.

Response: We have added the full names for the acronyms in lines 42, 63 and 65.

2. In the fourth paragraph, the three major complexes regulating apicobasal cell polarity are presented, CBR, PAR and SCRIB, but the rest of the manuscript focuses on PALS1, which is part of the CRB complex (acronym missing and to be added). Since PAR and SCRIB are not described, I think they should be removed to avoid confusion. I would rephrase into something like “One of the major complexes regulating apicobasal cell polarity is the the CRB complex...”.

Response: We have amended the text as suggested and added the full name for CRB in line 76.

3. In the fifth paragraph, the Crumbs PDM is introduced, but no description is provided. It might be worth adding a sentence to describe, with a comprehensive reference.

Response: We have added the following sentence and reference as suggested in line 89: “The CRB3 protein is the major crumbs isoforms expressed in human epithelial cells with the CRB3A splice form encoding the carboxy terminal PBM sequence ERLI that enables PALS1 binding (Margolis, 2018).”

4. A personal opinion more than a recommendation or a requested amendment, I think that a figure with a schematic of the different domains of SARS-CoV E protein, as well as the polarity complex, might help to visualise better the whole interactome presented in the manuscript and would really make the introduction outstanding.

Response: We have added schematics as suggested to a revised Figure 1.

5. In the first paragraph, the authors present the K_d measurements of interaction between PALS1 PDZ and SARS-CoV-1/2 E PBM peptides by ITC. Fig 1B present the ITC data, however the control titrations of the peptides into buffer are not reported. These should be added, either in the main figure or in a supplementary figure. This is particularly relevant because the heat of dilution observed at the end of the titration is quite high (almost half of the total signal recorded). This could be due to buffer mismatch (it is not clear from the Methods whether the peptides were resuspended in the very same protein buffer) or to some

contaminants of the peptides synthesis that are titrated, generating heat (this can happen with commercially synthesised material).

Response: We have added the relevant peptide into buffer titrations in an amended Figure 1. We also amended the methods in line 272 to specify that peptides were dissolved in water to prepare 5 mM stock solutions, which were subsequently diluted with buffer for the final experiment.

6. Also, the authors should report a table with averaged stoichiometry (n), ΔH , $-T\Delta S$ (\pm SD or SEM) obtained in the titrations. Ideally, considering that the PDZ and SARS-CoV-1/2 E PBM peptides complexes are very similar, similar enthalpy and entropy should be expected. If great differences were observed, the data should be discussed accordingly.

Response: We have added the thermodynamic parameters in a new Table 1.

	K_D (μ M)	ΔH (kcal/mol)	ΔG (kcal/mol)	$-T\Delta S$ (kcal/mol/K)	N
SARS-CoV-1 PBM	29.6 \pm 2.3	-0.67 \pm 0.01	-6.15 \pm 0.03	5.49 \pm 0.04	1.0 \pm 0.02
SARS-CoV-2 PBM	22.8 \pm 1.2	-0.68 \pm 0.01	-6.34 \pm 0.02	5.66 \pm 0.15	0.8 \pm 0.1
MERS-CoV PBM	NB	NB	NB	NB	NB

7. In the second paragraph, the authors introduce the structure of the complexes formed by PALS1 PDZ and SARS-CoV-1/2 E PBM peptides. The XRC structures show no folding of the small peptides. I was wondering whether the unstructured PBM peptides herein observed reflect previously reported structures of the SARS-CoV-1/2 E PBM (in greater constructs maybe) or PBM of other proteins. If the PBM is “usually” a structured element in greater constructs, the authors should discuss this. Another reason for investigating this more is that the high heat of dilution observed at the end of the ITC titration might be due to the transition of the peptides from (partially) folded to unfolded state during the binding. The analysis of the enthalpy and entropy obtained in the ITC experiments will guide this too.

Response: The sequences of both SARS-CoV-1/2 PBM adopt a beta-strand structure for the three C-terminal residues. We displayed the PBM peptide sequences as sticks, and in this presentation it is harder to see this. Our observations for the SARS-CoV peptides are similar to what we have previously observed for other PBM sequences such as beta-PIX, MCC or Vangl2 (see Lim et al JBC 2017, Caria et al FEBS J 2019 and How et al. Biochem J 2021). PBM sequences located at the C-terminus of proteins are typically unstructured, and fold upon binding to a PDZ domain binding groove, where they typically adopt a beta-strand like structure. Since our ITC analysis revealed an unfavourable entropic contribution, we speculate that the peptide is not moving from a folded to an unfolded state.

8. In paragraph 4, the authors indicate the amino acids of the SARS-CoV-1/2 E PBM peptides with a negative numbering. My understanding is that position 0 is fixed at V75 (SARS-CoV-1_E) and V76 (SARS-CoV-2_E). Not having the numbering anywhere stated is though confusing, so I would report the numbering somewhere, maybe at the top of the alignment presented in Fig 1A.

More on Fig 1A, the alignment reports a series of additional proteins that are never mentioned in the manuscript main text. I am assuming these are PBM of other proteins. The authors should clarify that. Finally, this alignment is a tad “amatorial”. A better alignment (with rigorous homology values) should be performed in Clustal (or similar), where it is possible to appreciate the conservation between PBM (all of those presented, not just the viral ones). A more detailed discussion on this figure should then be reported in the main text, since this is at the moment missing.

Response: We have made a number of changes to the alignment. The alignment was regenerated using Clustal Omega, and similarity assessment was performed using Jalview. We also provide detailed information regarding other sequences in the alignment, and have added the numbering scheme to assist with the examination of the structures. The alignment is now also called out in the section discussing the differing interactions of SARS-CoV-2 E and Crb with PALS1 PDZ in line 170.

9. In paragraph 4, the authors describe the role of F318 in the binding of CRB1 PBM to PALS1 and how this reflects on the binding SARS-CoV-1_E and SARS-CoV-2_E binding to PALS1 PDZ. How did the authors extrapolate the information about the rotation of F318 in the interaction with SARS-CoV-1/2_E? Was this obtained through alignment between complexes and apo-PALS1 PDZ? A more detailed discussion on this should be provided and a panel in Fig 3 focused on the description of this important point should be presented.

Response: We compared out structures of PALS1 PDZ with the apo-PALS1 PDZ structure from Ivanova et al (2015). In Ivanova et al, the role of F318 is discussed, based on a comparison of apo-PALS1 PDZ with the complex of PALS1 PDZ with Crumbs. We have generated an additional panel for Figure 3 to show this, and expanded the discussion around this point as follows: “A comparison of our structures with apo-PALS1 PDZ reveals that similarly, both SARS-CoV-1_E and SARS-CoV-2_E binding to PALS1 PDZ trigger rotation of PALS1 Phe318 out of the ligand binding groove to allow binding of SARS-CoV-1/2_E PBM (Fig. 3E).”

10. In paragraph 5, the fact that MERS-CoV E does not bind to PALS1 PDZ is rationalised through the fact that it presents a W where SARS-CoV-1/2_E have a V. Please, point the Trp in the alignment of Fig 1A to help the reader.

Response: We have marked up the Trp in the MERS-CoV E PBM using a red box in the revised alignment in colour to make it more visible for the reader.

11. Paragraph 2: In the sentence “Examination of both the PALS1 PDZ:SARS-CoV-1_E and PALS1 PDZ:SARS-CoV-2_E complex showed that the E protein PBM peptides are bound in the canonical ligand-binding groove formed by the β 2 strand and helix α 2 (Fig. 2).” ‘Complex’ should be amended into ‘complexes’.

Response: Changed as suggested in line 140.

12. The Fig 3B is described before Fig 3A. It might be worth swapping 3B and 3A figures to avoid confusion for the readers.

Response: We swapped panels 3A and B as suggested.

13. In paragraphs 3 and 4, the amino acids of SARS-CoV-1/2_E interacting with PALS1 PDZ are reported in “random” order. I understand that this follows the top-to-bottom representation in Fig 3, however, at this level it might be better to follow an alphanumerical order. I will leave this to the authors.

Response: The convention for PBM motif numbering is top-to-bottom, so we have left this as is.

14. A reference is missing for the sentence in paragraph 6 “A 6-mer peptide spanning the CRB1 PBM motif bound to PALS1 PDZ with an affinity of 16.8 μ M, comparable to the affinities we determined for our E protein PBM 8-mer peptides.”

Response: We have added the relevant reference to the statement in line 190.

15. In the sentence “As correct localisation of cell polarity proteins are essential for their function, mislocalisation of PALS1...”, ‘are’ should be corrected into ‘is’.

Response: Changed as suggested in line 200.

16. In the sentence “In summary, we have shown that SARS-CoV-1_E and SARS-CoV-2_E PBM bind to PALS1 PDZ with comparable affinities and establish the structural basis for CoV_E PBM binding to PALS1 PDZ. These findings provide a mechanistic basis for coronavirus induced perturbation of polarity signalling and may serve as a platform for the design of small molecule inhibitors to target the envelope protein interaction with PALS1 for antiviral therapy” there are two typos: ‘...comparable affinities, and established...’ and ‘...polarity signalling, and may...’.

Response: Changed as suggested in line 226.

Fig 1: Note that the space between values and units are missing.

Response: We have added a space between values and units.

17. Fig 2: I think it would be beneficial if this figure was reorganised so that panel C becomes panel A, since that the PALS PDZ : CRB1 complex was not characterised in this work. Also, it might be worth indicating in this complex the structural elements that are interacting with the E proteins, mentioned in the paragraph 3 of the Results section, to guide the reader through the results.

Response: We have reorganized Figure 2 as suggested, and the added labels to each secondary structure element now better supports the results section and discussion of the ligand binding groove.

18. Fig 3: Fig 3 reports 2 panels (D and E) that are never mentioned in the main text. These two panels should be moved to figure 2, since that they are actually reporting on the quality of the XRC electron density fitting.

Response: We move the panels as suggested, and now call out the moved panels in the main text in line 141.

19. I found Fig 3F quite busy, thus very difficult to follow. Few amendments are recommended. I would suggest focusing on the surface around V76 SARS-CoV-1/2_E only, since this is the object of the discussion here, and to align it with W81 of MERS-CoV_E to point at the different steric hindrance (I guess that the structure of MERS-CoV_E PBM could be easily modelled for this purpose). The surface is not necessary, so it could be probably hidden, so that the reader can focus on the V76/W81 vs PALS1 residues only. Also, there are way too many colours in this figure, making it quite complex to read. I would suggest to have each complex in the same colour, but two different shades for PALS1 and CoV_E protein respectively, so that it is easy to appreciate the interaction in each complex. Also, I believe that the SH3 domain of PALS-PDZ-SH3-MAGUK is not adding to this, but I will leave this to the author if they discuss better in the results the point of aligning this too. Finally, a minor amendment. The PALS-PDZ-SH3-MAGUK:Crumbs_PBM complex is referred as PALS-PDZ-SH3-GUK:Crumbs_PBM complex in the caption. I am not sure whether this is a typo, if so, please correct.

Response: We agree this panel is busy, but felt that it would still be a valuable addition to the manuscript. We have removed the surface as suggested, and now show the PALS1 chains from the different complexes in two different shades of pink, and the PBM peptides in two shades of cyan. We modelled the Trp81 from MERS CoV E and now show this as steel blue spheres. We have changed the figure legend to now state it is the MAGUK domain not GUK domain in line 566.

20. *E. coli* should be presented in italic. Note that some units are missing the space after the number (i.e. 3.5mg/mL instead of 3.5 mg/mL - this would apply to the whole manuscript).

Response: We amended the manuscript as suggested.

21. In the first paragraph, the authors use the words “Corona peptides”. A personal opinion, more than a recommendation, descriptions such as “Coronavirus peptides” or “viral peptides” sound more appropriate.

Response: We amended the manuscript and now use coronavirus peptides.

22. Still in the first paragraph, the authors describe the full set of plates tested in automated crystallisation trials. I think this is redundant, since that the successful crystallisation conditions are stated clearly in the third paragraph. I would suggest this is removed to avoid confusion for the reader. At the same time, if the authors observed crystals in different conditions tested and feel like sharing this information with the community, then they can provide more details.

Response: We have removed the redundant information as suggested. We sadly only observed one crystallization condition that featured protein crystals.

Reviewer #2:

1. In the abstract sentence that reports the binding affinities (lines 32-33), the protein that was tested is not named (SARS-CoV1 and 2 E protein).

Response: We amended the manuscript and now explicitly state that we examined SARS-CoV-1 and -2 E proteins in line 33.

2. Extension of 'Fig. 1' to Fig 1A, for example, may be helpful throughout the manuscript. Where this is done, panel B is often mentioned in the text before panel A. Perhaps panels A and B should be switched to show SARS-CoV1 results before SARS-CoV2 if they are referred to in that order.

Response: We have amended Figure call outs to be more detailed, and reorganized Figure 1.

3. Isothermal titration calorimetry experiments seem to not have been taken fully to completion. To improve this, additional points using both less and more ligand could allow a complete sigmoidal curve to be generated. However, if this has proved challenging, the data remain valid as all points appear close to the fitted curve. This is supported by the binding affinity of PALS1 PDZ:SARS-CoV2 E presented here being close to that previously reported (although SARS-CoV1 E is quite different but this is addressed in the main text).

Response: We were unable to obtain a complete sigmoidal curve from a single experiment owing to the difficulty of an optimal protein:peptide concentration ratio to get sufficient datapoints at both ends of the curve. We did not feel comfortable combining datapoints from different experiments in a single curve for display as a Figure, so have left the ITC curves as per initial submission.

4. The authors refer to certain secondary structure features (i.e. beta strands) although these are not annotated on any of the figures. The addition of a panel showing these (or annotation of a current figure) may aid the reader in identifying the regions being referred to in the text, although not essential.

Response: We have annotated the secondary structure elements on the cartoon structures in Figure 2.

5. On line 124, the authors state that there is no significant difference between PALS1 PDZ:SARS-CoV2 E and PALS1 PDZ:Crumbs although figure 2E shows quite a large difference in the lower region of the E/Crums overlay. Perhaps the text should be altered to specify no significant difference in the upper/central regions or ‘most’ of the peptides.

Response: This was clearly somewhat ambiguously phrased. We intended for this statement to apply to the PALS1 PDZ1 domains, not the E peptide. We have amended this as follows in line 144:” There was no significant difference between PALS1 PDZ domains from the PDZ:SARS-CoV-2_E complex and PALS1 PDZ from an endogenous complex PALS1 PDZ:Crumbs (PDB ID 4UU5), which were superimposed to produce an overall root-mean-square deviation (RMSD) of 0.80 Å (Fig. 2, E). “

6. The abbreviation of human T lymphotrophic virus-1 is missing in line 183.

Response: we have added the abbreviation for human T lymphotropic virus-1 in line 217.

7. The sentence starting on line 210 doesn't make sense. Perhaps 'in which' needs re-wording on line 212?

Response: we have split this sentence to improve clarity in line 245.

8. What is the composition of buffer A? First mentioned in line 214.

Response: we now specify buffer A early in the protein purification section as follows in line. 239: “Cells were harvested by centrifugation at 4000 g for 20 minutes using the JLA 9.1000 rotor (Beckman Coulter Avanti J-E), then re-suspended in 100 ml in buffer A (50 mM Tris pH 8.0, 150 mM NaCl, 1mm Ethylenediaminetetraacetic acid (EDTA)) and homogenized using an Avestin EmulsiFlex homogenizer.”

9. Line 218 and 245: missing word 'concentrators' or similar after '...molecular weight cut off...'. I.e. '...molecular weight cut off' what?

Response: we amended the sentence in line 252 to: “100 ml of buffer A was then added to collect cleaved protein, with the flowthrough concentrated at 4°C using 3 kDa molecular weight cut-off centrifugal concentrator (Amicon® Ultra 15 Millipore) with a benchtop centrifuge (Allegra X-15, Beckman Coulter) at 4000g.”

10. Table showing data/resolution of the structures is presented., however, these are not mentioned in the text or figure legends.

Response: we now call out Table 2 at the start of the results and discussion section in line 134 as follows: “To understand the structural basis of the PALS1 PDZ and SARS-CoV-1/2 E PBM interaction, we determined the crystal structures of PALS1 PDZ bound to the SARS-CoV-1_E (SRVPDLLV) and SARS-CoV-2_E (EGVPDLLV) to resolutions of 1.74 and 1.9 Å, respectively (Table 1).”

11. Figure 2 legend: proteins mentioned in A and B are the wrong way around compared to the labels in the figure panels. I.e. Legend states A shows SARS-CoV1 E although the figure is labelled as SARS-CoV2 E. The final line of the legend is also confusing and potentially contains an error in line 467 where the Crumbs peptide is labelled as both cyan and forest green (with SARS-CoV2 E in green).

Response: We have corrected the legend, and clarified the colouring for the Crumbs peptide in line 540.

12. Figure 3 legend is very repetitive and could be re-worded to be more succinct.

Response: We consolidated the section describing panels 3D and 3E, and moved them to Figure 2 as per request from reviewer 1. We also consolidated the text for the remaining panels to make the legend less repetitive.

13. Do panels 3D and E add anything to the figure? May be easier to see if shown without the surface of PALS1 PDZ. Panel F is good in that it shows the amino acid labels although appears very busy and difficult to interpret. Again, this may benefit from the removal of the surface view of PALS1 PDZ.

Response: We moved panels 3D and E to Figure 2, as requested by reviewer 1. We have also removed the surface from 3F (which is now 3E).

Reviewers' Comments:

Reviewer #1:

Remarks to the Author:

I am overall very satisfied by the amendments and the additional data and clarifications provided by Javorsky and colleagues regarding the manuscript entitled 'Structural Basis of Coronavirus E protein interactions with human PALS1 PDZ domain'.

Amendments on all sections are satisfactory and fully met my requests reported in the previous round of review. I am particularly pleased by the additions on the ITC data and the improvements in the figures. I am also very pleased with the discussion provided in the rebuttal for my questions.

Having said that, I strongly recommend the publications of this manuscript, upon very minor amendments of few typos encountered:

1. In line 272, I would specify that "peptides were subsequently diluted with buffer for the final experiment", to provide full clarity on the experimental procedure.
2. Fig 1D caption does not clearly states that the bottom panels present the ligand-into-buffer controls. This should be added to avoid confusion for those readers that are not familiar with the ITC technique.
3. The reference to the ITC experiments in the main text (line 126) is reported as Fig 1C, but it should be amended into Fig 1D.
4. Fig 2C caption now describes what is now reported in Fig 1A, so this should be amended accordingly.
5. In line 560, A) should be amended into A,B), since it describes both panel A and B of Fig 3.

I would like to thank the authors for their contribution to the COVID19 field and wish them and their families health and a quick return to normality in these trying times.

Kind regards and stay safe.

Reviewer #2:

Remarks to the Author:

Authors of the manuscript entitled 'Structural basis of Coronavirus E protein interactions with human PALS1 PDZ domain' have addressed all comments raised by both reviewers in the first round of review.

All textual issues have been corrected and comments of figure clarity have been addressed by adding new labels/panels and/or removing the surface view to improve visibility where suggested.

Response to reviewer comments

Reviewer #1 (Remarks to the Author):

1. In line 272, I would specify that “peptides were subsequently diluted with buffer for the final experiment”, to provide full clarity on the experimental procedure.

Response: We now state: “Peptides were dissolved in water to prepare 5 mM stock solutions, and subsequently diluted with buffer for the final experiment.”

2. Fig 1D caption does not clearly states that the bottom panels present the ligand-into-buffer controls. This should be added to avoid confusion for those readers that are not familiar with the ITC technique.

Response: This is an excellent point, we have now added the following statement in the caption for Figure 1D: “Top panels show peptide into protein titrations, bottom panels show peptide into buffer control titrations. “

3. The reference to the ITC experiments in the main text (line 126) is reported as Fig 1C, but it should be amended into Fig 1D.

Response: We have corrected this.

4. Fig 2C caption now describes what is now reported in Fig 1A, so this should be amended accordingly.

Response: We have corrected this to ensure the panel order is consistent with the legend.

5. In line 560, A) should be amended into A,B), since it describes both panel A and B of Fig 3.

Response: We have corrected this and now state: “Detailed view of interactions in the binding groove of PALS1 PDZ in complex with **(A, B)** SARS-CoV-2 E protein PBM peptide (green) or SARS-CoV-1 E protein PBM peptide (orange) or **(C)** Crumbs PBM peptide (cyan) (PDB ID 4UU5)²⁹.”

Reviewer #2 (Remarks to the Author):

No response required.